DeepBindPoc: a deep learning method to rank ligand binding pockets using molecular vector representation

Zhang Haiping 1
http://orcid.org/0000-0002-5541-234X Saravanan Konda Mani 1
Lin Jinzhi 1
http://orcid.org/0000-0002-6103-0700 Liao Linbu 2
Ng Justin Tze-Yang 3
http://orcid.org/0000-0002-6564-5059 Zhou Jiaxiu 4 shirleyzjx@163.com
Wei Yanjie 1 yj.wei@siat.ac.cn
1 Center for High Performance Computing, Joint Engineering Research Center for Health Big Data Intelligent Analysis Technology, Shenzhen Institutes of Advanced Technology, Chinese Academy of Sciences , Shenzhen, Guangdong Province , China
2 College of Software Technology, Zhejiang University , Zhejiang Province, Zhejiang , China
3 School of Biological Sciences, Nanyang Technological University , Singapore , Singapore
4 Shenzhen Children’s Hospital , Shenzhen, Guangdong Province , China
de Azevedo Walter Jr
Electronic publication date: 2020 Apr 6
Publication date: 2020
Volume: 8
Electronic Location ID: e8864
Received 2019 Dec 13; Accepted 2020 Mar 8
Copyright: © 2020 Zhang et al.
Copyright year: 2020
Copyright holder: Zhang et al.
License: This is an open access article distributed under the terms of the Creative Commons Attribution License, which permits unrestricted use, distribution, reproduction and adaptation in any medium and for any purpose provided that it is properly attributed. For attribution, the original author(s), title, publication source (PeerJ) and either DOI or URL of the article must be cited.
License URL: https://creativecommons.org/licenses/by/4.0/

Keywords: Ligand pocket identification, Deep neural network, Mol2vec, Densely fully connected neural network, Protein–ligand interactions

Funding: National Key Research and Development Program of China 2018YFB0204403 and 2016YFB0201305 Shenzhen Basic Research Fund JCYJ20180507182818013, GGFW2017073114031767 and JCYJ20170413093358429 National Science Foundation of China under U1435215 and 61433012 National Natural Youth Science Foundation of China 31601028 China Postdoctoral Science Foundation 2019M653132 CAS Key Lab 2011DP173015 Shenzhen Discipline Construction Project for Urban Computing and Data Intelligence Youth Innovation Promotion Association This work was supported by the National Key Research and Development Program of China under grant Nos. 2018YFB0204403 and 2016YFB0201305, the Shenzhen Basic Research Fund under grant no. JCYJ20180507182818013, GGFW2017073114031767 and JCYJ20170413093358429, National Science Foundation of China under grant nos. U1435215 and 61433012; the National Natural Youth Science Foundation of China (grant no. 31601028), the China Postdoctoral Science Foundation (grant no. 2019M653132), CAS Key Lab under grant no. 2011DP173015. This work was also supported by the Shenzhen Discipline Construction Project for Urban Computing and Data Intelligence, Youth Innovation Promotion Association, CAS to Yanjie Wei. The funders had no role in study design, data collection and analysis, decision to publish, or preparation of the manuscript.

==============================
Accurate identification of ligand-binding pockets in a protein is important for structure-based drug design. In recent years, several deep learning models were developed to learn important physical–chemical and spatial information to predict ligand-binding pockets in a protein. However, ranking the native ligand binding pockets from a pool of predicted pockets is still a hard task for computational molecular biologists using a single web-based tool. Hence, we believe, by using closer to real application data set as training and by providing ligand information, an enhanced model to identify accurate pockets can be obtained. In this article, we propose a new deep learning method called DeepBindPoc for identifying and ranking ligand-binding pockets in proteins. The model is built by using information about the binding pocket and associated ligand. We take advantage of the mol2vec tool to represent both the given ligand and pocket as vectors to construct a densely fully connected layer model. During the training, important features for pocket-ligand binding are automatically extracted and high-level information is preserved appropriately. DeepBindPoc demonstrated a strong complementary advantage for the detection of native-like pockets when combined with traditional popular methods, such as fpocket and P2Rank. The proposed method is extensively tested and validated with standard procedures on multiple datasets, including a dataset with G-protein Coupled receptors. The systematic testing and validation of our method suggest that DeepBindPoc is a valuable tool to rank near-native pockets for theoretically modeled protein with unknown experimental active site but have known ligand. The DeepBindPoc model described in this article is available at GitHub (https://github.com/haiping1010/DeepBindPoc) and the webserver is available at (http://cbblab.siat.ac.cn/DeepBindPoc/index.php).

Introduction

A protein can interact specifically with binding partners such as small molecules, nucleic acids or with other proteins in the cell to perform its different important biological functions. Understanding how and where these molecules bind in the protein targets provides valuable information for therapeutic design because it is essential to mimic or enhance a function in the cell (Lionta et al., 2014). Predicting ligand binding pockets in proteins is one of the key issues in the early stages of structure-based drug discovery and still an unresolved problem in computer-aided drug design (Liang, Edelsbrunner & Woodward, 1998; Miller & Dill, 2008). The accurate specification of ligand binding pockets affects the efficiency of the whole computational drug discovery process (Glaser et al., 2006). Several ligand binding pocket predictors have been developed in the past decades using various approaches like physico-chemical based, geometric based and machine learning based methods, respectively (Stank et al., 2016). A geometric based pocket identification method is introduced by Kuntz et al. (1982) in the early 1980s. Following this, a similar method “SURFNET” is proposed by another group (Laskowski, 1995). LIGSITE and Pocket picker search for cavities from the atomic coordinates of proteins by mapping the interface as grids and spheres (Hendlich, Rippmann & Barnickel, 1997; Laurie & Jackson, 2005; Weisel, Proschak & Schneider, 2007). Also, many methods use evolutionary information to predict pockets, because the protein sequences diverge during the evolution (Schelling, Hopf & Rost, 2018). Fpocket, concavity, and CASTp are hybrid methods which use similarity searches from existing databases and other geometric indices to identify pockets (Capra et al., 2009; Le Guilloux, Schmidtke & Tuffery, 2009; Tian et al., 2018).

A recently developed P2Rank, a machine learning based tool, demonstrates a strong prediction ability for protein pocket (Krivák & Hoksza, 2018). There are several deep learning-based methods to identify native pockets (Jiménez et al., 2017; Pu et al., 2019). However, most current available methods (Saberi Fathi & Tuszynski, 2014; Jiménez et al., 2017; Krivák & Hoksza, 2018; Pu et al., 2019) have not incorporated ligand information in pocket identification, indicating these methods would have serious limitations in pocket which induces changes in protein structure upon ligand binding. Some machine learning models have an over fitting problem which only performs well when a test case is close to the training data but fail to predict better on additional independent data (Ursenbach et al., 2019). Also, most current models training based on data considers the native conformations as positive conformations. Such training datasets are different compared to the real application data, which are usually derived from prediction. Often, such models have limitations for applications in real drug-discovery situations where the potential pocket was generated by the software and only near-native pocket exist.

In principle, the comparison and classification issues can be effectively addressed by using advanced deep learning methods. The Dense Fully Connected Neural Network (DFCNN) is one such deep learning algorithm which is very suitable for comparison and classification for protein ligand complexes (Ragoza et al., 2017). In our previous work, we used DFCNN to perform the classification of ligand-binding residues and non-ligand binding residues accurately (Zhang et al., 2019a, 2019b). In the present work, we rescore and rank ligand binding pockets accurately by deep learning of interface contact information in the form of vectors. Our model also resolves the discrepancy between training and real applications by treating the near-native pocket as training positive. Additionally, ligand information is also added to enhance model accuracy and generalization. Through this article, we strongly believe that the way we used molecular vector representation of proteins and the ligand information in the model would help to perform enhanced prediction to rank ligand-binding pockets than other existing traditional methods.

Materials and Methods

Data sets and decoys generation

The PDBbind v2017 forms the source of our present study and it contains aver 16,151 Protein–ligand complexes (Wang et al., 2005). Around 270 complexes that contain rarely occurring atom types (such as) that could not be processed by RDKit was removed (Wildman & Crippen, 1999) and after cleaning the data, we have 14,491 protein–ligand complexes for further processing. The fpocket tool was used to generate the pockets for the given protein with the default parameters (Le Guilloux, Schmidtke & Tuffery, 2009). The pockets and their corresponding ligands were used to generate the input for our model. The data preparation process is similar to our previous work (Zhang et al., 2019a). The ligands were converted into SMILES format by open babel (O’Boyle et al., 2011) and then converted into a 300-dimensional vector by mol2vec tool (Jaeger, Fulle & Turk, 2018). The basic idea of mol2vec is to consider the SMILES string as molecular sentence which are composed of words (substructure), and like the natural language processing method word2vec, an unsupervised machine learning method was used to construct the mol2vec by learning vector of each word based on a large amount of available chemical compounds dataset (corpus) (Krallinger et al., 2015). The residues in the pocket are converted into a 300-dimensional vector by mol2vec tool and summed up into a 300-dimensional vector to represent the pocket. The protein–ligand binding is then represented as a 600-dimension vector which concatenates the ligand vector and pocket vector by an in-house python script. The three-dimensional structures shown in the article are plotted by using Visual Molecular Dynamics (VMD) and Chimera (Humphrey, Dalke & Schulten, 1996; Pettersen et al., 2004).

Positive and negative datasets

We constructed two positive datasets with different strategies. Strategy 1 is by defining a collection of atoms that fall within 1 nm around the known ligand as the known pocket. The potential pockets that are close to the known pockets (have C alpha center distance with known pocket smaller than 0.3 nm) were taken as the positive dataset. Strategy 2 is that known pockets are chosen directly as the positive dataset. The negative dataset is constructed from fpocket predictions on the potential pockets of the proteins in the PDBbind database. The parameters are set to default to perform fpocket prediction. We randomly choose three pockets that have C alpha center distance with a native pocket larger than 1 nm. If the number of pockets that have larger than 1 nm distance to the native pocket is less than three, we consider all the pockets. The selected predicted pockets together with the ligands are taken as the negative dataset. If the decoy pocket is far away from the known ligand binding pocket (center of distance between a known pocket and decoy pocket are larger than 3 nm) and the pocket’s vector is not similar to that known ligand binding pocket, we assume the pocket is not the near-native pocket of ligand. We understand that this is a big approximation, we can’t guarantee the defined non-ligand binding pocket is not a druggable site, but it is highly possible that defined non-ligand binding pocket was not the given ligand’s binding site. The deep learning can tolerate noise (very small portions of unreliable data) and such approximation still can be used in the construction of our model.

To make sure each vector of decoy pocket is far away from their corresponding known pocket; we have used the following formula to calculate their vector similarity. Using the cutoff value of 0.995, we remove those pockets that are highly similar to the native one.

(1) Sij=(Vi∗Vj)/(|Vi|∗|Vj|)

where the Sij is the measured similarity between pocket i and pocket j, the Vi is the vector of pocket i, and the Vj is the vector of pocket j. The dataset was divided into two groups as near natives as positive “A” and native pockets as positive “B”. The training, validation, testing for two groups of datasets is shown in Fig. 1. The model generated with training A is validated and tested with dataset B and vise versa.

Figure 1 Classification of datasets into different groups.

The ET_A, ET_B, ET_C stands for extra test A, extra test B and extra test C, respectively.

Preparation of extra test sets

We have collected protein–ligand complexes that are deposited in the PDB database after the year 2018 (Berman et al., 2000). We remove redundancy by only keeping one PDB structure if the structures are from the same gene. These protein–ligand complexes are not in the PDBbind 2018 dataset (Wang et al., 2005) and used as an extra testing set. The extra testing set is further divided into three parts: 11 cases that fpocket have generated near-native pockets (extra test set A); 6 cases that fpocket have not generated near-native pockets (extra test set B); 2 cases have two pockets corresponding to different ligands (extra test set C). The classification and grouping of data are presented in Fig. 1. The 6 cases that fpocket can’t generate near-native decoys were used the native pocket as positive. The proteins with PDB identifier 6QTN (A, F chain) and 5ZG2 contain two pockets with different ligands bound. We attempted to check whether our method can successfully identify correct pockets for each of the ligands. The known pocket and near-native pocket are defined by the same method as above. The proteins in the extra test set A and B were subjected to the prediction by the P2Rank with its default parameters for comparison (Krivák & Hoksza, 2018; Jendele et al., 2019).

Preparation of G-protein coupled receptor independent test dataset

We retrieved 98 GPCR-ligand complexes from the GPCRDB database (https://www.gpcrdb.org/) (Pándy-Szekeres et al., 2018). The near-native pocket was defined as the same as the previously mentioned procedure. We are interested to test whether our methods can perform well on those challenging GPCR proteins, characteristic of a true structure-based drug discovery application scenario.

Construction of deep learning model

The details of the model construction procedure are illustrated in Fig. 2. It contains data processing, model training, validation and testing. We use the DFCNN inspired by DenseNet as our model (Huang et al., 2017). The DFCNN model architecture is similar to our previous work (Zhang et al., 2019a). The fully connected neural network is suitable for vectors as inputs. Moreover, DenseNet can overcome the gradient vanishing problem and allows many deep layers for learning more abstract features. This model has shown good performance in identifying protein–ligand binding affinity in our previous work (Zhang et al., 2019a). It has advantages in protein–ligand binding estimation over most other machine learning methods including Support Vector Machine (Suykens & Vandewalle, 1999), RandomForest (Breiman, 2001), XGBoost (Chen & Guestrin, 2016), Convolutional Neural Network (CNN) (Krizhevsky, Sutskever & Hinton, 2012). Densely fully-connected neural network (DFCNN) and CNN were built using Keras (Chollet, 2015) with Tensorflow back end (Abadi et al., 1983). The DFCNN has 16 densely connected layers outputting 100 units simultaneously plus a normal fully-connected layer outputting one unit as the final output. Specifically, densely connected layer refers to a layer taking all outputs of its preceding layers as its input which can remarkably solve the problem of gradient vanishing. Rates for dropout layers are all set to 0.25. The dense layers employed ReLU activation function except for the output layers which employed sigmoid activation function. Input for the network has been normalized to make its mean and standard deviation to be 0 and 1 separately. The Adam optimizer was used to minimize the binary cross-entropy of DFCNN.

Figure 2 The workflow of DeepBindPoc model.

Data normalization and performance evaluation

All the data have been normalized before the final input for the model. The normalization is as follows: (2) t=data_set

(3) tnormalize=(t−mean)/std

where t is the data set value, the mean is the data mean value and the std is the data standard deviation. We have tested two normalization strategies, one is based on fixed mean and standard deviation values directly from the training dataset (mean = −0.5696 and std = 30.8744 for using near-native as positive and mean = −0.9610 and std = 63.6607 for using native as positive). The other strategy involves normalizing by dataset itself, which is used for comparison in the present study. We find normalize by training dataset is more reliable, so we used normalized by training dataset unless specifically stated. Several metrics were used to evaluate the proposed models, including accuracy, Area Under the receiver operating characteristic Curve (AUC) (Hanley & McNeil, 1982), Matthews Correlation Coefficient (MCC), True Positive Rate (TPR), specificity and sensitivity.

Web server

The protein structures and its known ligands in PDB format are required as input to the webserver http://cbblab.siat.ac.cn/DeepBindPoc/index.php. We first use the fpocket to generate the pocket decoys and we use mol2vec to convert the pocket and ligand into the vectors. After the protein and ligand vectors are concatenated, DeepBindPoc will score decoys with native like possibility, and select the top three decoys as the potential pockets. The predicted pocket name was shown on the page along with the fpocket score and DeepBindPoc score. The results can also be downloaded as a file for the convenience of the user. We also provide the batch mode to the user and provide a zip file of proteins with their corresponding ligands. For both the single model and the batch model, we have provided an example input for the convenience of the user.

Results

DeepBindPoc performance on the training, validation and testing datasets

To determine the hyperparameter of epoch number, we check the convergences by monitoring the change of accuracy and loss value in both the training and validation process with the increasing epoch number. The results are shown in Fig. S1. The performance of the validation set has converged at epoch 1,500. The AUC, accuracy, precision and MCC were used as evaluation metrics. DeepBindPoc model is generated based on training A of strategy 1 (described in method section). DeepBindPoc’s performance on the training A, validation A, test A, training B, validation B and test B set is shown in Table 1. The results are normalized by training data. The results reveal that training A has AUC value of 0.9972, the accuracy of 0.98 and an MCC value of 0.95. In the case of validation A, the DeepBindPoc achieves AUC value of 0.98, the accuracy of 0.93 and 0.86 MCC. Although our test A has an unbalanced positive–negative proportion (677 positives to 5,822 negative), our model still achieves AUC value of 0.98, the accuracy of 0.95, and an MCC of 0.77. Other performance indicators (TPR, precision) also support the good performance of our model among training, validation and test dataset. The training data have TPR value of 0.97 and a precision of 0.98. The validation and test have a TPR value of 0.9 and 0.89 and a precision of 0.96 and 0.71, respectively.

Table 1 The DeepBindPoc performance on Training A, Validation A, Test A, Training B, Validation B and Test B.

The normalization strategy is based on the Training dataset. The DeepBindPoc is trained by Training A. The details of each data set were described in Materials and methods section. Pos_size and Neg_size in the table denotes size of the positive and negative dataset.

Data set	AUC	Accuracy	TPR	Precision	MCC	Pos_size	Neg_size	
Training A	1.00	0.98	0.97	0.98	0.95	6,000 × 3	18,000	
Validation A	0.98	0.93	0.90	0.96	0.86	1,000	1,000	
Test A	0.98	0.95	0.89	0.71	0.77	677	5,822	
Training B	1.00	0.98	0.98	0.98	0.96	6,000 × 3	18,000	
Validation B	0.99	0.97	0.99	0.96	0.94	1,000	1,000	
Test B	1.00	0.97	0.98	0.97	0.94	7,491	5,822	

When using the native as positive data for testing (Training B, Validation B and Test B in Table 1), DeepBindPoc has comparable or better performance. This indicates that our model is reliable under different situations and our model’s performance can be further improved by generating high quality near-native decoys. To demonstrate the advantage of near-native as training positive, we generated a DeepBindPoc_native model based on training B of strategy 2 for comparison (Table S1). The model trained by near-native pocket as positive data (DeepBindPoc, the result shown in Table 1) demonstrated better performance than the model trained by native pocket as positive data (DeepBindPoc_native, the result shown in Table S1). The DeepBindPoc can perform well for both native and near-native as positive. The DeepBindPoc_native performs well only for native as the positive, while its performance on Training A, Validation A and Test A (which have the near-native as positive) have decreased significantly. The possible reason is that when training directly by the native pocket as positive, it is difficult for the model to generalize the near-native as positive data due to feature distribution inconsistency between training and usage. It should be noted that in the real application scenarios, the native (conformation) is often unavailable. Usually, only near-native mixed with nonnative (conformation) can be generated by computational tools such as fpocket. The DeepBindPoc trained by positive data strategy 1 in the method section is more reliable (Table 1). The DeepBindPoc_native performance on the training A, validation A and test A are not as good as in Table 1, indicating its limitations in real application scenarios.

The DeepBindPoc performance on nonredundant 2019 new dataset

The performance of the extra testing set was very close to the real application because all the pockets are generated from fpocket prediction and near-native pocket is selected from the decoys. The performance of extra test A is shown in Table 2. It is observed that the normalized strategy based on training set has relatively better performance over the normalized strategy based on data itself for most of the performance indicators except TPR. The model has achieved AUC value of 0.93, the accuracy of 0.88 and MCC of 0.38 respectively. Furthermore, the TPR (0.75) and precision (0.24) are also high. This indicates the model has high potential to identify the near-native pockets correctly from a bunch of decoys generated by fpocket. The comparison of the performance of DeepBindPoc and fpocket based on whether they can successfully identify near-native pocket within top 5, 3 and 1, respectively is presented in Table 3. Our method is strongly complementary to those traditional methods where prediction is only based on physico-chemical property of pocket. Because we currently haven’t developed a method to generate pocket decoys, we depend on fpocket or other methods such as p2rank to first generate the pocket decoys, then we can do the rescoring. However, it is still possible in the future to develop a method that can iteratively generate almost all possible pocket decoys, and then using our method to do the rescore. In this way, our method may more easy to use and better to compare the performance with other software. Our method has incorporated the information of ligands together with the pockets, and the training is over a large data set with feature distribution close to real application scenarios.

Table 2 The DeepBindPoc performance on the extra test set A and the independent GPCR dataset, which is close to the real application.

Pos_size and Neg_size in the table denotes size of the positive and negative dataset.

Data set	Normalized strategy	AUC	Accuracy	TPR	Precision	MCC	Pos_size	Neg_size	
Extra test A	1#	0.90	0.80	0.83	0.18	0.32	12	238	
Extra test A	2#	0.93	0.88	0.75	0.24	0.38	12	238	
GPCR set	1#	0.96	0.85	0.95	0.16	0.36	98	3,050	
GPCR set	2#	0.97	0.91	0.93	0.26	0.46	98	3,050	
Note:

#1, based on data itself; #2, based on training set.

Table 3 The comparison of the performance of DeepBindPoc and fpocket based on whether they can successful identify near-native pocket with in top 5, 3 and 1, respectively.

PDB ID	DeepBindpoc	Fpocket	
In top 5	In top 3	In top 1	In top 5	In top 3	In top 1	
6NQ0	×	×	×	×	×	×	
6J4H	✓	✓	×	×	×	×	
6J0O	✓	✓	✓	✓	✓	×	
6IEZ	✓	✓	✓	×	×	×	
6I2A	✓	✓	✓	×	×	×	
6GGG	✓	✓	×	✓	✓	✓	
6K04	✓	✓	✓	✓	×	×	
6GEV	✓	✓	×	✓	✓	✓	
6E3T	✓	×	×	×	×	×	
6PSJ	✓	✓	✓	✓	✓	✓	
6SJM	✓	✓	✓	✓	✓	✓	
5OVE	✓	✓	×	×	×	×	
Summary	11	10	6	6	5	4	

There are 10 out of 11 and 6 out of 11 cases have the native predicted in top five by the DeepBindPoc and fpocket, respectively. Only the 6NQ0 have failed in the top five of DeepBindPoc, but it is still better ranked than fpocket (ranked 7 vs 34, the yellow pocket shown in Fig. S2). There are 11 out of 12 and 7 out of 11 cases have the native predicted in top three by the DeepBindPoc and fpocket, respectively. In 10 out of 12 proteins, there is successful identification of the correct pocket within the top three predictions, among which 5 cases have the correct pocket at the top prediction. We also observe two cases that failed to correctly identify within the top three predictions by visualizing the spatial relationship between the known pocket and the predicted pockets. Interestingly, we found even in failed predicted cases, our method often predicted better than the fpocket rank (Fig. S2). For example, in the case 6E3T, the third predicted pocket is close to the native pocket and shares several critical residues for the ligand binding. The fourth predicted pocket is the near-native pocket which is still better compared to the fpocket rank 9. In another case 6NQ0, the near-native pocket was ranked 7th by DeepBindPoc, while ranked 34th by fpocket. Table 3 presents the prediction by DeepBindPoc with the training dataset (fixed mean and standard deviation) as a normalization strategy.

The DeepBindPoc performance on five cases that fpocket have failed to generate near-native pockets

We have selected five proteins deposited in PDB since 2019 for extra testing. The fpocket fails to generate near-native conformation for them. By carefully examining these cases, it is found that sometimes, fpocket can only generate short fragments of the native pocket. Hence, we choose the known pocket as positive to do the testing. The results of normalization by training A are shown in Table 4. With the known pocket as the positive, all the known pockets of each 6 cases are ranked 1st, indicating our method is extremely accurate once the known pocket is within the predicted decoys. This also indicates, our DeepBindPoc model is promising to increase prediction accuracy by incorporating other software that can extract closer to native pocket decoys than the fpocket software.

Table 4 The top three predicted values of DeepBindPoc on the five cases (extra test B) that fpocket failed to generate near-native; the corresponding fpocket prediction value was also given for comparison.

Interestingly, for each case, the native pockets are all ranked as top 1. The Normalized strategy is based on the training set. The pocket is Native pocket or pocket from the fpocket generation. Pos and Neg in the table denotes positive and negative data. The number after “poc” in table represents rank of the prediction.

Pocket	Prediction	Label	
6MT8_native_poc	1.00	pos	
6MT8_poc1	1.00	neg	
6MT8_poc9	0.98	neg	
6J8V_native_poc	1.00	pos	
6J8V_poc31	1.00	neg	
6J8V_poc18	1.00	neg	
6J5W_native_poc	1.00	pos	
6J5W_poc74	1.00	neg	
6J5W_poc42	1.00	neg	
6I53_native_poc	0.99	pos	
6I53_poc3	0.98	neg	
6I53_poc1	0.85	neg	
6F83_native_poc	1.00	pos	
6F83_poc4	0.09	neg	
6F83_poc3	0.05	neg	

The P2Rank performance on the extra test set

The performance of P2Rank on the 17 proteins of the extra testing set was shown in Table 5. The P2Rank have demonstrated highly accurate prediction. There are 10 cases with near-native pocket and impressively the near-native pocket for each case is ranked top. However, there are 7 cases that the near-native pockets are not in the predicted pocket decoys. Since the P2Rank is based on a decision tree algorithm, inconsistent performance for some cases is observed and this may be due to overfitting, which is a common problem in the decision tree-based model. By incorporating fpocket, P2Rank and our DeepBindPoc together, it is hopeful to further improve the accuracy of pocket identification. There are 4 cases that both the fpocket and P2Rank haven’t generated near native pocket (6F83, 6I53, 6J5W, 6J8V), which indicates the pocket generation methods should also be improved. The DeepBindPoc perform better than both P2Rank and fpocket in three cases, which is shown in Fig. 3. The pockets generated by P2Rank for three proteins (6I2A, 6F83 and 6J5W) failed. The native pocket of 6I2A is correctly ranked as 1st by DeepBindPoc, while the fpocket score ranked as 17th (Fig. 3A). Only a small part of the native pocket of 6F83 have generated by fpocket, the fpocket ranked it in 4th, while our DeepBindPoc successfully ranked it as 1st (Fig. 3B). Two parts of the native pocket of 6J5W are generated by fpocket, the DeepBindPoc ranks the larger parts 13th, while the fpocket ranks it 91th (Fig. 3C). If we add the native pocket in the prediction, DeepBindPoc can always correctly identify the correct ones (Table 4), it indicates that the DeepBindPoc or its further updated version, have potential to guide an accurate generation of pocket decoys.

Table 5 The performance of P2Rank on the 17 proteins of the extra testing set A and B.

It was observed at least one near-native pocket among all the prediction, the near native pocket often ranks top (in this test all are rank top 1). However, there are still some cases where the near-native is not in the predicted pocket decoys.

Protein	Number of predicted Pocket	Near-native pocket	
5OVE	9	None	
6J4H	31	6J4H_poc1	
6E3T	21	6E3T_poc1	
6K04	1	6K04_poc1	
6F83	3	None	
6GEV	5	6GEV_poc1	
6PSJ	5	6PSJ_poc1	
6GGG	9	6GGG_poc1	
6I2A	7	None	
6I53	6	None	
6IEZ	5	6IEZ_poc1	
6J0O	10	6J0O_poc1	
6J5W	45	None	
6SJM	3	6SJM_poc1	
6J8V	27	None	
6MT8	7	6MT8_poc1	
6NQ0	26	None	

Figure 3 The three cases that have better performance over both P2Rank and fpocket.

The Protein Data Bank identifiers of three cases (6I2A, 6F83 and 6J5W) are shown as (A), (B) and (C), respectively. The figure is plotted by VMD.

The DeepBindPoc performance on two cases which have two pockets corresponding to two different ligands

The A and F chain of 6QTN have two pockets, each corresponding to a different known ligand (ACF and GTP, Fig. 4A). The 5ZG2 also contains two pockets, each corresponding to a known ligand (Fig. 4B). We attempted to test our model in these challenging scenarios to evaluate whether our model has the potential to identify the ligand-specific pockets. The results reveal that our model has identified the pocket in 6QTN for the ACF in the 8th prediction while the fpocket has rank 35th (Table S2; Figs. 4A and 4B, normalized by training A). The model has successfully ranked the correct pocket for the GTP in the second position, but in this case, fpocket is better, which ranks this pocket as top 1. The results reveal that our model has found the pocket in 5ZG2 for the ZKI in the 2nd rank while the fpocket prediction ranked 19th and the pocket for the 9C0 is the 2nd prediction while the fpocket prediction ranked 13th (Table S2; Fig. 4). These observations suggest that our model can find novel pockets for a known active ligand by incorporating the traditional fpocket method.

Figure 4 The two-ligand corresponding to two pocket cases.

(A) Two pockets of the protein with PDB ID: 6QTN. (B) Two pockets of the protein with PDB ID: 5ZG2. The figure is plotted by UCSF chimera.

Performance on the GPCR membrane protein dataset

The GPCR pocket may have quite different physico-chemical features as the non-membrane proteins due to the lipid environment, which may pose extra challenges for pocket identification. The performance of DeepBindPoc on membrane proteins was shown in Table 2 (normalized by training A). Our model is not so accurate in such challenging cases that have a well-known difficulty in obtaining membrane protein structures experimentally. The reason for the inaccurate performance is due to the following reasons: (1) most of the training data is not membrane protein, so the training data and the membrane protein–ligand testing set may have some different feature distribution; (2) the membrane proteins contain lipids, and often involve conformation change during ligand binding and the physico-chemical and geometric feature of GPCR pocket is different; (3) the 3D structure of ligand binding with membrane protein sometimes is not that much reliable; (4) the accessibility of ligand binding to the GPCR pocket is different compared to soluble proteins. In the future, we want to use the transfer learning technique to develop a GPCR specific pocket prediction model, which will first train on a generalized dataset and then retrain the model’s top layers on the membrane protein–ligand dataset.

Discussion

Similarity and dissimilarity comparisons of physico-chemical properties of protein structures will help to gain valuable insights to develop methods to accurately predict important regions in the proteins and have implications in computer-aided drug design (Wei, Nadler & Hansmann, 2007; Saravanan & Selvaraj, 2013; Zhang et al., 2016). In this article, we present an accurate pocket identification method “DeepBindPoc” by incorporating the ligand information and the model’s usage has shown in Fig. S3. We demonstrated that our model has a complementary advantage over the popular fpocket score in many test cases. Our method takes advantage of the current popular word2vec representation to generate protein pocket and ligand representation that can accurately include critical physico-chemical properties. Because of the simple data structure of a vector, the model is quite efficient during prediction. As shown in our previous work, the densely fully connected neural network is suitable to preserve the vector’s complete information compared to CNN. Another advantage is its many deep layers allow it to learn more abstract and high-level features underlying large data. The systematic validation of DeepBindPoc implies the efficiency of the model in estimating the ligand binding site of the protein. Our pocket identifying method also has the potential to be implemented in some specific protein classes, such as GPCR proteins, which contain the most important drug therapeutic targets. Moreover, their structure and exact drug binding site are much harder to obtain by the experimental methods, such as X-ray Crystallography.

Conclusion

Through the article, we show that DeepBindPoc can be used in complement with computational methods such as, fpocket to accurately identify the drug binding site, which is very crucial in the first stage of GPCR-related drug development. In a real application, it is worth trying several different pocket generation tools to comprehensive access the potential pockets. This indicates the importance of developing a new pocket generation tool based on the novel scoring function such as DeepBindPoc or its future updated version. Further, it is possible to combine spatial information with the molecular vector by convolutional graphic neural network in the near future. Although our model is not an ultimate solution for the pocket identification problem, we provide a choice of coupling other traditional methods and provide helpful insights for future development of powerful pocket identification tools.

Supplemental Information

Supplemental Information 1 Table S1.

The performance of DeepBindPoc_native model which training by native pocket as positive (Training B). The normalization strategy is based on Training dataset. The Training B, Validation B, and Testing B below are using native as positive. The Training A, Validation A, and Test A are the generated dataset with the near-native as positive dataset.

Click here for additional data file.

Supplemental Information 2 Table S2.

Demonstrate the predicted value of DeepBindPoc on the decoys of extra test C (6QTN and 5ZG2). The fpocket prediction value was given for comparison. The normalized strategy is based on the training set. The pocket_A and pocket_B are top predicted pocket of 6QTN by DeepBindPoc score for GTP and ACP, respectively. The pocket_C and pocket_D are the top predicted pockets of 5ZG2 by DeepBindPoc score for ZKI and 9C0, respectively.

Click here for additional data file.

Supplemental Information 3 The convergences check by monitoring the loss and accuracy of the training and validation set.

Click here for additional data file.

Supplemental Information 4 Analysis of the four cases that failed to predict correct pocket in top 3 predictions in table 3 (mark yellow).

Some of the false-positive pocket in extra data set predicted by DeepBindPoc is still near the native.

Click here for additional data file.

Supplemental Information 5 The usage of the DeepBindPoc.

Click here for additional data file.

Additional Information and Declarations

Competing Interests

Author Contributions

Data Availability

The authors declare that they have no competing interests.

Haiping Zhang conceived and designed the experiments, performed the experiments, analyzed the data, prepared figures and/or tables, authored or reviewed drafts of the paper, and approved the final draft.

Konda Mani Saravanan conceived and designed the experiments, performed the experiments, analyzed the data, prepared figures and/or tables, authored or reviewed drafts of the paper, and approved the final draft.

Jinzhi Lin analyzed the data, prepared figures and/or tables, and approved the final draft.

Linbu Liao conceived and designed the experiments, performed the experiments, analyzed the data, prepared figures and/or tables, and approved the final draft.

Justin Tze-Yang Ng conceived and designed the experiments, prepared figures and/or tables, and approved the final draft.

Jiaxiu Zhou conceived and designed the experiments, analyzed the data, prepared figures and/or tables, authored or reviewed drafts of the paper, and approved the final draft.

Yanjie Wei conceived and designed the experiments, analyzed the data, prepared figures and/or tables, authored or reviewed drafts of the paper, and approved the final draft.

The following information was supplied regarding data availability:

Data is available at GitHub: https://github.com/haiping1010/DeepBindPoc.

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
