# Peer review of "DeepBindPoc: a deep learning method to rank ligand binding pockets using molecular vector representation"

_PeerJ, doi:10.7717/peerj.8864_

## Round 0.1 · original submission · Major Revisions

Three specialists in the field evaluated this submission. Two of them have major concerns related to your manuscript. I agree with them.

Reviewer 1 ·

Basic reporting

The publication requires significant revisions in grammar and overall use of the English language. Almost every sentence from the 'Introduction' section on, presents an issue with number concord or is missing a determinant. Some examples are:
Line 85: "Our method is not mutually exclusive with those traditional method that include spatial information."
The line after, line 86, reads:
"Since the way we used molecular vector have ignore the spatial information..."
In some cases, the grammar errors make it hard to understand the actual meaning of the text.

Experimental design

The construction of the datasets needs some clarifications. Besides that, it is nowhere explained in the text why some pockets from Fpocket are included in the negative set. The fact that there is not a ligand present in that pocket, does not mean that the pocket is druggable.
Overall, the methods section needs some further explanations on how the datasets were created and the role of Fpocket on each of the steps.

It is also somewhat confusing, that Fpocket is used to generate the datasets which are then used for the training, but then the performance of DeepBindPoc is precisely compared against Fpocket.

Validity of the findings

The idea of the work is the implementation of a machine learning protocol to detect the pocket in a protein for a given ligand. The input set contains over 16000 proteins from the PDBbind database. It is, however, unclear how many were rejected due to RDkit parsing issues and how many proteins ended up on each dataset. The construction of the datasets needs further explanations, it is unclear for the reader to see where the datasets from Strategy 1 and 2 ended up in Tables 1-3. The comparison against other programs such as Fpocket, also needs further information. Table 3 lists some pdbs and compares the ability to find the correct pocket in the first 5,3 and first guesses. It is however nowhere stated why these PDBs where selected.

Overall, the publication needs major revisions and I do not favour its publication at this stage.

Additional comments

The implementation of an algorithm (DeepBindPoc) that detects binding pockets for a given ligand in a protein has tremendous applications in the early drug discovery pipeline. The fact that the program has been made available as a web server is also a very nice feature. However, the paper as it is in its current state needs major revisions and should be rewritten. The fact that the neural network is trained on a set of pockets obtained with Fpocket, but then later Fpocket's performance is compared against DeepBindPoc does not sound like a rigorous method or should be better explained.
The method section needs a better explanation of how the datasets were created, and why they were created this way. This process should be better reflected in Figure 1, and the Strategies used for the creation should either be better explained or integrated later in the Results tables.
Lastly, the use of the English language needs a major revision. Almost all sentences present a grammar error.

Reviewer 2 ·

Basic reporting

In this paper, authors have proposed a new deep learning method, DeepBindPoc for identifying and ranking ligand-binding pockets in proteins. They built DeepBindPoc by utilizing information about the binding pocket and corresponding ligand. The authors have extensively tested and validated the model with standard procedures on multiple datasets, including a dataset with G-protein Coupled receptors. The article is well written and relevant references were provided. Hypotheses were backed by data. Overall, I consider this work very important in the field of structure-based drug design. I recommend this article for publication in PeerJ.


















0.+

Experimental design

No comment

Validity of the findings

No commnent

Additional comments

1. The performance of DeepBindPoc on membrane proteins was not that good. What could be the possible reason? How do you want to address it in the near future?

2. line 87, page 2: "have ignore....". it should be "have ignored".

3. there are article mistakes.. for eg. line 88-90: "To sum up, this method has strong complementary advantage over existing traditional methods, because we incorporated ligand information and effective molecular vector representation in our model". It should be "......has a strong....... "

·

Basic reporting

The authors introduce a deep-learning-based tool DeepBindPoc for the identification of binding sites. The model is built using both information about the binding site and associated ligand using the mol2vec tool. The model was trained on structures in PDBIND 2018 and tested and validated on PDB structures which were deposited in 2019.

I reviewed basically the same text about a half a year ago for the Bioinformatics Journal. Although the authors have addressed some of the major issues remain.

* The paper needs to be proofread by a native speaker. For example:
* Our method is not mutual exclusive with those traditional method that includes spatial information. ---> Our method is not mutually exclusive with those traditional methods that include spatial information.
* Since the way we used molecular vector have ignore the spatial information, ... ---> Does not make sense at all
* As the point of the method is to do rescoring of existing predictions with the ligand information present, I believe it should be also compared with rescoring using protein-ligand scoring functions available in existing structure-based drug design tools. There even exist deep-learning-based approaches such as https://www.ncbi.nlm.nih.gov/pmc/articles/PMC5479431/ (the virtual screening scenario).

Experimental design

* Common issue with the evaluation of ML methods is information leakage. I strongly believe this is exactly the case here. It is very probable, that the train and test sets contain virtually the same structures because no effort what so ever was put into filtering out homologs. The least one should do is to use sequence identity to filter out similar sequences. And even then, since we are not interested here in the whole structures, but only in active sites it is quite possible that there can be almost identical binding sites in both test and train sets since the structure similarity, we are interested in is only local. So to really be sure there is no information leakage, one should use some local structure similarity measure.

* The AUC and other measures are computed based on whether a true binding site was found among the top 3 predicted pockets (that is three selected from Fpocket results). However, one should also show the baseline, that is what are the numbers for Fpocket only to see what numbers DeepBindPoc actually tries to improve. I.e. the authors should pick top 3 pockets from Fpocket and show the numbers for that.

* At one point authors compare to P2Rank. When looking into P2Rank paper, one can see that the authors have chosen Top-N+k evaluation strategy. That is if the protein has N pockets and you choose k=0 you look at top N pockets. Since here the authors search for binding sites of a specific ligand, the N is always 1 and k = 2. To paint the full picture, the authors should also provide at least the Top1 and Top2 or TopN in general, where N is the number of predicted pockets given by Fpocket.

* I don't really see the point of testing on the 6 cases where FPocket failed to generate the pocket since in real-world scenario, DeepBindPoc wouldn't have anything to rescore.

* Related to the previous point - I tried to pick one of the pockets (6mt8) which FPocket failed to predict and run the prediction with [2] and the correct pocket was found as the first one. This might show that choosing Fpocket as the underlying main pocket detection tool might have been not the best choice and the authors should really see how DeepBindPoc would improve on state of the art approaches.

Validity of the findings

See the previous section.

---

## Round 0.2 · Minor Revisions

In the abstract, the authors say, "important physic-chemical" it should be "important physical-chemical". Please correct it.

·

Basic reporting

My comments have been addressed.

Experimental design

My comments have been addressed.

Validity of the findings

My comments have been addressed.

Additional comments

My comments have been addressed.

---

## Round 0.3 · accepted · Accept

The authors carried out all modifications indicated by this editor and by all reviewers. In my view, the manuscript can be accepted as it is.